# COVID-19 Vaccine Effectiveness and Risk Factors of Booster Failure in 480,000 Patients with Diabetes Mellitus: A Population-Based Cohort Study

**DOI:** 10.3390/microorganisms13050979

**Published:** 2025-04-24

**Authors:** Maria Christina L. Oliveira, Daniella R. Martelli, Ana Cristina Simões e Silva, Cristiane S. Dias, Lilian M. Diniz, Enrico A. Colosimo, Clara C. Pinhati, Stella C. Galante, Fernanda N. Duelis, Laura E. Carvalho, Laura G. Coelho, Maria Eduarda T. Bernardes, Hercílio Martelli-Júnior, Fabrício Emanuel S. de Oliveira, Robert H. Mak, Eduardo A. Oliveira

**Affiliations:** 1Department of Pediatrics, Health Sciences Postgraduate Program, School of Medicine, Federal University of Minas Gerais (UFMG), R. Engenheiro Amaro Lanari 389/501, Belo Horizonte 30310-580, MG, Brazil; chrismariana@gmail.com (M.C.L.O.); acssilva@hotmail.com (A.C.S.e.S.); profacristianedias@gmail.com (C.S.D.); lilianmodiniz@gmail.com (L.M.D.); claracpinhati@gmail.com (C.C.P.); stellacgalante@gmail.com (S.C.G.); duelisfernanda@gmail.com (F.N.D.); lauracarvalhoe@gmail.com (L.E.C.); coelholaura26@gmail.com (L.G.C.); dudatbernardes@gmail.com (M.E.T.B.); 2Health Science/Primary Care Postgraduate Program, State University of Montes Claros (Unimontes), Montes Claros 39401-089, MG, Brazil; daniellareismartelli@yahoo.com.br (D.R.M.); hmjunior2000@yahoo.com (H.M.-J.); fabricioemanuel1@hotmail.com (F.E.S.d.O.); 3Department of Statistics, Federal University of Minas Gerais (UFMG), Belo Horizonte 30310-580, MG, Brazil; enricoc57@gmail.com; 4Division of Pediatric Nephrology, Rady Children’s Hospital, University of California San Diego, La Jolla, CA 92093, USA; romak@health.ucsd.edu

**Keywords:** COVID-19, diabetes mellitus, outcomes, vaccine, effectiveness

## Abstract

To investigate the real-world effectiveness of COVID-19 vaccines in a large cohort of patients with diabetes mellitus (DM), we analyzed all >18-year-old patients with COVID-19 registered in a Brazilian nationwide surveillance database between February 2020 and February 2023. The primary outcome of interest was vaccine effectiveness against death, evaluated using multivariate logistic regression models. Among the 2,131,089 patients registered in the SIVEP-Gripe, 482,677 (22.6%) had DM. After adjusting for covariates, patients with DM had a higher risk of death than those without comorbidities (adjusted odds ratio [aOR] = 1.43, 95% CI, 1.39–1.47). For patients without comorbidities (72.7%, 95% CI, 70.5–74.7) and those with DM (73.4%, 95% CI, 68.2–76.7), vaccine effectiveness was similar after the booster dose. However, it was reduced in patients with DM associated with other comorbidities (60.5%; 95% CI, 57.5–63.2). The strongest factor associated with booster failure was the omicron variant (aOR = 27.8, 95% CI, 19.9–40.1). Our study revealed that COVID-19 vaccines provided robust protection against death in individuals with DM. However, our findings underscore the need to update vaccines and develop tailored strategies for individuals with diabetes, especially those with additional underlying conditions.

## 1. Introduction

Four years have passed since the World Health Organization (WHO) declared COVID-19 as a global pandemic on 11 March 2020. Early investigations revealed that the elderly and individuals with pre-existing chronic conditions were at significantly higher risk of severe COVID-19 illness and death due to SARS-CoV-2 infection [1,2,3,4].

Diabetes mellitus (DM) is a major public health concern affecting millions of people globally [5,6]. The COVID-19 pandemic has highlighted the vulnerability of individuals with DM, with studies reporting high rates of severe illness and mortality among this population [7,8,9,10]. A comprehensive study has shown that individuals with diabetes accounted for nearly 10% of severe cases and 17% of deaths worldwide [11]. Individuals with diabetes mellitus (DM) are at an elevated risk of severe COVID-19 owing to a confluence of factors, including immune dysfunction [12], chronic inflammation [13], and metabolic dysregulation [14]. Furthermore, DM exacerbates COVID-19-related morbidity and mortality, potentially extending the recovery duration [15,16]. Vaccines help mitigate these risks by inducing both humoral and cellular immune responses, which are essential for preventing severe disease [17]. As a result, many countries have prioritized patients with DM for COVID-19 vaccination [18]. However, research suggests that these individuals may have a weaker immune response to the vaccine than those without DM [19,20]. Furthermore, vaccine hesitancy has been observed among people with diabetes, often due to misinformation, lack of information, and mistrust [21].

Even after the pandemic control, SARS-CoV-2 may persist as an endemic seasonal pathogen for several years, evolving over time [22,23]. It remains uncertain how effective COVID-19 vaccinations have been in patients with potentially weakened immune systems due to pre-existing chronic conditions [24]. Although vaccination has been instrumental in mitigating the pandemic, real-world data on COVID-19 vaccine efficacy in people with DM remain limited [25]. The primary objective of this retrospective population-based cohort study was to investigate vaccine effectiveness (VE) against COVID-19-related death in a large group of patients with DM.

## 2. Materials and Methods

Data Source and Study Design. We conducted a population-based, retrospective cohort analysis, including all patients who were admitted to hospitals and registered in the Influenza Epidemiological Surveillance Information System (SIVEP-Gripe). SIVEP-Gripe is a comprehensive national database in Brazil that tracks cases of severe acute respiratory infection (SARI) across the country [26,27]. The reporting of SARI is mandatory in Brazil and the database collects data from patients admitted to both public and private hospitals. The detailed information about the SIVEP-Gripe database, including the reporting form and data dictionary, coding, and de-identified data, is publicly available at https://opendatasus.saude.gov.br/dataset accessed on 20 February 2023. Additional information is provided in the Appendix A. The study followed the Strengthening the Reporting of Observational Studies in Epidemiology (STROBE) guidelines for observational cohort studies [28].

Participants. The inclusion criterion was patients aged ≥ 18 years with laboratory-confirmed COVID-19 who were registered in the database between February 2020 and February 2023. All patients included had a positive quantitative reverse transcription polymerase chain reaction (RT-qPCR) test or antigen test results for SARS-CoV-2. The flowchart displays the complete information about the included and excluded cases (Figure 1).

Exposure of interest. The primary exposure of interest was DM. Patients with DM were identified in the SIVEP-Gripe database by retrieving data from specific fields concerning pre-existing chronic medical conditions. For analysis purposes, we divided the cohort into four groups, according to the presence of DM and other comorbidities: Group 1, patients without comorbidities; Group 2, patients with non-DM comorbidities; Group 3, patients with DM without other comorbidities; and Group 4, patients with DM associated with other comorbidities. 

Covariates. This study included a range of demographic and clinical covariates to account for potential confounding variables. Demographic data included age, sex, ethnicity, educational attainment, and geographic macro-region. Age was categorized into four strata: 19–29 years, 30–59 years, 60–79 years, and >80 years. Ethnicity data were based on the Brazilian Institute of Geography and Statistics (IBGE) classification system, encompassing White, Black, Brown, Asian, and Indigenous categories. Brazil’s geopolitical macro-regions (north, northeast, central–west, southeast, and south) were included as covariates because of potential historical, social, economic, and healthcare system variations. Educational attainment was categorized into five levels, based on the highest achieved school degree: illiterate, elementary school, middle school, high school, and college. Clinical covariates included the date of birth, symptom onset date, and admission date. Additionally, signs and symptoms at presentation (fever, cough, respiratory distress, gastrointestinal symptoms, and oxygen desaturation), nosocomial infections, and pre-existing comorbidities were documented. Oxygen saturation obtained at admission was classified as a dichotomous variable in the SIVEP-Gripe database (cut-off point of <95%). In addition to DM, the dataset provides data on other comorbidities including asthma, obesity, immune deficiencies; malignancies; and heart, lung, kidney, neurological, and hematological diseases. For analysis purposes, the presence of comorbidities was categorized into four levels based on the number of pre-existing conditions identified (none, one, two, and three or more).

Vaccination Status. Vaccination status was categorized at the time of the onset of the symptoms as unvaccinated, partially vaccinated (one dose), fully vaccinated (two doses) and boosted (three doses). In Brazil, the Ministry of Health (MS) is the only provider of COVID-19 vaccines. Four vaccine schedules of different platforms have been authorized in Brazil for the adult population: mRNA vaccine (Pfizer, BioNTech), inactivated virus vaccine (Sinovac; CoronaVac), and viral vector vaccines (ChAdOx1 nCoV-19, Astrazeneca, and Ad26.COV2.S, Johnson & Johnson’s Janssen). Additional information on the Brazilian vaccination program and about the vaccination status of the individuals is provided in the Appendix A.

Outcomes. The primary outcome was VE against death. Additionally, this study assessed in-hospital mortality and healthcare resource utilization, including admission at intensive care unit (ICU) and the use of respiratory support among hospitalized patients. Respiratory support was categorized as none, non-invasive, and mechanical ventilation.

Statistical analysis. The analysis was performed in four steps. The first stage involved a comparative analysis of the demographic, clinical, and epidemiological characteristics of the four groups included in the analysis. Descriptive data for all eligible patients were presented as means and standard deviations, medians and interquartile ranges, or counts and proportions, as appropriate. For intergroup comparisons, Student’s *t*-test and Pearson’s chi-square test were used, where applicable. Second, to assess the effect of DM and other comorbidities on the survival of patients with COVID-19, we performed a binary logistic regression analysis. The model was controlled for age, sex, ethnicity, geographic macro-region, oxygen saturation at admission, nosocomial infection, epidemiological week of admission, viral strain, and number doses of vaccine. We also assessed the survival analyses by comparing the cumulative incidence function (CIF) of death among groups. Mortality was evaluated by competing risks analysis, using cumulative incidence function (CIF). Discharge was analyzed as a competing event by competing risks analysis. [29]. Third, we compared vaccine effectiveness against death conferred by the vaccines among the groups. For this analysis, we developed five models using the binary regression logistic method, including one overall model for the entire cohort and one model for each group included in the analysis. In the models, death was included as a dependent variable, and vaccination status, stratified according to the number of doses, was included as an independent variable. All models were adjusted for age, sex, ethnic group, geographic macro-regions, educational level, viral strain, and epidemiological week of admission. The results were expressed as VE (%) using the formula 100 × (1–adjusted odds ratio) and their 95% confidence intervals (CI). Finally, we evaluated potential factors associated with vaccine failure in this cohort. We defined vaccine failure as death after the patient received three doses of vaccine. For this analysis, we developed a binary regression logistic model with death as the dependent variable and clinical and epidemiological factors as the independent variables (age, sex, ethnicity, macro-regions, educational level, viral strain, and epidemiological week).

## 3. Results

### 3.1. Baseline Demographic and Clinical Characteristics

The cohort comprised 2,131,089 patients, including 482,677 patients with diabetes (22.6%). For the analysis, we divided the cohort into four groups: 911,270 (42.8%) patients without comorbidities, 737,142 (34.6%) patients with non-DM comorbidities, 381,828 (17.9%) patients with DM and other pre-existing conditions, and 100,849 (4.7%) patients with DM without associated comorbidities. The demographic and clinical characteristics of the cohort according to the four groups are shown in Table 1. In general, patients with DM were significantly older at presentation. Proportionally, patients with diabetes were more frequently from the south and southeast regions, of White ethnicity, and had a lower educational level than those without comorbidities. These patients also presented with a greater proportion of respiratory distress and oxygen saturation of <95% at baseline than patients without comorbidities. Cardiovascular disease (71.8%), hypertension (27.8%), obesity (15.7%), neurological disorders (8.6%), and kidney disorders (7.5%) were the most prevalent comorbidities.

### 3.2. Outcomes

Regarding the overall clinical outcomes, 689,380/1,836,099 patients (37.5%) were admitted to the ICU; 1,053,002/1,809,738 (58.2%) required non-invasive oxygen support; and 369,811/1,809,738 (20.4%) required mechanical ventilation. The overall in-hospital mortality rate was 32% (681,681/2,131,089). The clinical outcomes according to group are shown in Table 1. The prevalence of DM was 27.5% among patients admitted in ICU, 29.3% in those requiring mechanical ventilation, and 29.1% among deceased patients. Patients with DM had a higher prevalence of ICU admission (44% vs. 35.6%), mechanical ventilation (25.4% vs. 18.9%), and death (41% vs. 29.3%) than those without comorbidities.

The estimated probabilities of fatal outcomes in the first 30 days of hospitalization were 30.3% and 40.7% for the non-diabetic and diabetic cohorts, respectively. The CIF of death according to the groups is shown in Figure 2. The estimated probabilities of fatal outcomes within the initial 30-day period of hospitalization were 23.6%, 33.6%, 38.3%, and 43.7% for patients without comorbidities, those with DM, those with other comorbidities excluding DM, and those with both DM and additional comorbidities, respectively.

The results of the univariate binary logistic regression analysis for the risk factors associated with mortality in the entire cohort indicated that all groups of patients with comorbidities were consistently associated with an increased risk of death compared with the groups without comorbidities (Appendix A). After adjusting for potential clinical and epidemiological confounders, all patient groups demonstrated a significantly higher risk of mortality than those without comorbidities: diabetes without other comorbidities (OR = 1.43, 95%CI, 1.39–1.47), other comorbidities without DM (OR = 1.87, 95%CI, 1.85–1.90), and DM with other comorbidities (OR = 2.18, 95%CI, 2.14–2.21) (Figure 3).

### 3.3. Vaccine Effectivennes

Among the 1,887,890 hospitalized patients with a known vaccination status, a significant majority (78.3%) were not fully vaccinated. This included 1,479,074 individuals who had received no doses, 103,774 who had received one dose, and 199,526 who had received two doses. Only 105,516 patients were fully vaccinated with all three doses at the time of data collection. Remarkably, 581,375 (96.1%) of the 605,164 patients who died in this cohort were not fully vaccinated. Similarly, among the 198,116 patients with DM who died, a similar proportion (96.5%) were not fully vaccinated.

Figure 4 shows the vaccine effectiveness (VE) against death for each group included in the analysis according to the number of received doses. Two or three doses of the vaccine conferred significant protection against death in all the groups. However, after the booster dose, there was a noticeable increase in vaccine protection in all the groups. Overall, VE against death was 67.2% (95% CI: 66.0–68.4) after the booster dose. Notably, after three doses, VE for patients without comorbidities (72.7%, 95%CI, 70.5–74.7) and patients with DM (73.4%, 95%CI, 68.2–76.7) were similar, with overlapping confidence intervals. Interestingly, VE against death was reduced for patients with other comorbidities (58.6%, 95%CI, 56.4–60.7) and patients with DM and other comorbidities (60.5%, 95%CI, 57.5–63.2) albeit still statistically significant. Again, there was an overlap between the confidence intervals for the former two groups.

### 3.4. Booster Failure

Among the 105,516 patients who received three doses of the vaccine, 23,789 (22.5%) had a fatal outcome. After adjustment for multivariate regression analysis, the following factors were significantly associated with booster failure in protection against death: diabetes associated with other comorbidities (but not in cases of diabetes without comorbidities), age (on an incrementally increasing gradient for older individuals), male sex, Brown or Black ethnicity, and low educational attainment (also in a stepped gradient) (Figure 5). However, the strongest factor associated with booster failure was the virus variants. Patients hospitalized during the delta (OR = 5.3, 95%CI, 3.8–7.3) and omicron (OR = 27.9, 95%CI, 16.9–45.8) waves had a remarkably increased risk of booster failure (Figure 5).

## 4. Discussion

Key points. This large cohort study examined COVID-19 outcomes and VE in a population of patients with DM hospitalized during the three-year pandemic period in Brazil. Our findings confirmed the heightened risk of severe COVID-19 and mortality among patients with DM compared to those without comorbidities, even after adjusting for confounders. Importantly, we observed comparable protection against death from COVID-19 conferred by the three vaccine doses in both DM patients and those without comorbidities. However, VE was reduced in patients with DM associated with additional comorbidities, although significant protection against death persisted. Notably, we observed a high rate of vaccine failure in patients hospitalized during the omicron wave.

Comparative analysis: clinical outcomes. Our findings underscore the strong association between diabetes mellitus (DM) and severe COVID-19 outcomes, consistent with previous studies [1,30,31,32,33,34]. This alignment is further supported by two recent large-scale meta-analyses [11,35]. In our cohort of hospitalized COVID-19 patients, the prevalence of DM was substantial (22.6%), escalating to 27.3% in ICU admissions, 29.3% in those requiring mechanical ventilation, and 29.1% among deceased patients. These findings mirror those of the recent meta-analysis by Li et al. [11], which demonstrated a progressive increase in DM prevalence with increasing COVID-19 severity. Our adjusted analysis revealed a significantly higher risk of death for patients with DM (OR 1.43, 95% CI 1.39–1.47). This risk was amplified in DM patients with other comorbidities (OR 2.18, 95% CI 2.14–2.21). These results corroborate those of a meta-analysis by Li et al. [35], reporting a pooled adjusted risk ratio of 1.43 for DM-associated mortality. In addition, Schlesinger et al. [36] highlighted worsened COVID-19 prognosis in individuals with severe DM and other comorbidities.

In the second phase of our study, we identified the risk factors for COVID-19 mortality in individuals with DM. After adjusting for potential confounders, age (in a stepped gradient), male sex, Brown, Black, and Indigenous ethnicities, patients from the northern region, pre-existing comorbidities (also in an incremental stepped gradient), low oxygen saturation at admission, and nosocomial infection were associated with increased mortality risk. Conversely, higher education and COVID-19 vaccination (two or three doses) were protective factors. Our findings align with previous research, demonstrating that age, male sex, and comorbidities are general risk factors for COVID-19 mortality in patients with DM [37,38,39,40]. The finding of an increased risk of in-hospital mortality in patients with diabetes of non-White ethnicity in our cohort is consistent with previous studies of the general adult population hospitalized with COVID-19 in Brazil [41,42]. As in many countries, ethnicity in Brazil is also strongly linked to the social determinants of health, including socioeconomic disparities, housing, education, income, and access to quality healthcare [26,43].

Comparative analysis: vaccine effectiveness. Previous research has indicated that individuals with DM may have reduced vaccine efficacy, including seasonal influenza [44,45]. Consequently, the effectiveness of COVID-19 vaccines in this population has been questioned [46]. A recent systematic review provides an overview of the COVID-19 VE in individuals with and without DM [25]. Overall, for all outcomes (infection, symptomatic illness, hospitalization, or death), VE data were numerically lower for individuals with DM than for controls, but there were several overlapping CIs [25]. Nevertheless, the included studies were severely heterogeneous, precluding meta-analysis [25]. In an assessment of COVID-19 VE against acute symptomatic SARS-CoV-2 infection in patients with comorbidities in the UK, Whitaker et al. [47] reported a VE of 43.2% (95% CI, 26.0–56.3) after the first dose and 72.9% (95% CI, 25.8–90.1) after the second dose in individuals with DM [47]. Our findings show that COVID-19 vaccination significantly reduced the mortality risk in individuals with DM, with protection increasing from 43.2% after the second dose to 73.4% after the booster. Notably, DM patients without comorbidities responded similarly to non-DM patients hospitalized with COVID-19, whereas those with additional comorbidities had a reduced but still significant vaccine benefit. Limited real-world data exist regarding COVID-19 vaccine effectiveness (VE) in patients with DM. Our results are consistent with those of a large Hungarian study by Molnár et al. [48], which reported similar VE in DM patients and controls, with higher effectiveness in older DM patients receiving mRNA vaccines. For non-mRNA vaccines, VE was comparable to our findings (approximately 50–70%) [48].

Finally, we assessed the risk factors for booster failure among individuals who received a third dose but subsequently died of COVID-19 during hospitalization. Consistent with the general population, older age and male sex were associated with an increased risk of death after booster vaccination. Although diabetes alone was not a risk factor, it significantly increased the risk of booster failure when combined with other comorbidities (61% increase in risk). Our study revealed a heightened risk of booster failure following the emergence of SARS-CoV-2 variants, particularly during the omicron period. This aligns with previous research demonstrating decreased vaccine effectiveness owing to waning immunity and the ability of the virus to evade immune responses [49,50,51]. In particular, the omicron variant significantly reduced the protection provided by the primary vaccine against severe disease compared to pre-omicron strains [52]. As a result, many countries have implemented booster programs to safeguard vulnerable populations, successfully increasing protection against severe omicron disease [53,54]. Nevertheless, SARS-CoV-2 continues to evolve, with omicron sub-lineages emerging, and some studies have shown that neutralizing antibody titers declined rapidly within three months of the third dose, particularly against sub-lineages BA.2.12.1 and BA.4/5 [55]. Taken together, these findings underscore the need for continued research and adaptive vaccination strategies, especially for high-risk individuals [53,54].

Limitations and Strengths. Our study has several limitations. First, the SIVEP-Gripe dataset’s limited comorbidity information, restricted to binary (yes/no) responses without detailed clinical characteristics, hinders a more comprehensive analysis. Consequently, we were unable to incorporate crucial variables, such as diabetes phenotypes and glycemic parameters, limiting our ability to thoroughly investigate the risk factors for the studied outcomes. Furthermore, it is important to acknowledge that the administrative nature of the database used in this study precludes a more detailed analysis, such as the assessment of the initial lung injury or the determination of the clinical management of patients. Second, given the focus on hospitalized patients, our findings may not be generalizable to all individuals with DM and SARS-CoV-2 infections. Third, the lack of individual-level vaccination data, including the timing between doses, vaccine types, and antibody responses, has limited our capacity to investigate important issues such as waning immunity or vaccine-specific effectiveness. Nevertheless, the strength of this study lies in its large cohort size, enabling a comprehensive examination of clinical characteristics, risk factors, outcomes, and vaccine responses among hospitalized patients with diabetes and COVID-19 in Brazil during the pandemic.

Public Health Implications and Future Directions. From a public health perspective, our findings highlight the need for targeted vaccination strategies to ensure adequate protection for individuals with DM. Additionally, strategies to promote and maintain healthy lifestyles should be prioritized at the population level, as they may help reduce the burden of COVID-19 among patients with chronic diseases [56]. Further studies exploring innovative methodologies, such as machine learning techniques, are essential to improving risk stratification and developing tailored prevention strategies at the population level [57].

## 5. Conclusions

In summary, our analysis of a large national database of hospitalized patients with confirmed SARS-CoV-2 infection revealed that individuals with DM experienced a more severe COVID-19 illness and a higher mortality risk than those without DM. Our study underscores the substantial protective effect of the COVID-19 booster dose against mortality in hospitalized patients with DM. Notably, VE in DM patients without comorbidities was comparable to that in healthy individuals. However, the reduced effectiveness in patients with multiple comorbidities suggests potential biological differences in the immune response, which warrant further investigation. Furthermore, we observed a concerning rate of booster failure in preventing death among patients hospitalized during the delta and omicron waves. The emergence of viral variants appears to be the most critical determinant of vaccine effectiveness. Given the heightened risks faced by individuals with DM, targeted vaccination strategies are crucial. Additional studies are needed to establish an optimal vaccination schedule for patients with DM and to determine the role of updated vaccines in years to come.

## Figures and Tables

**Figure 1 microorganisms-13-00979-f001:**
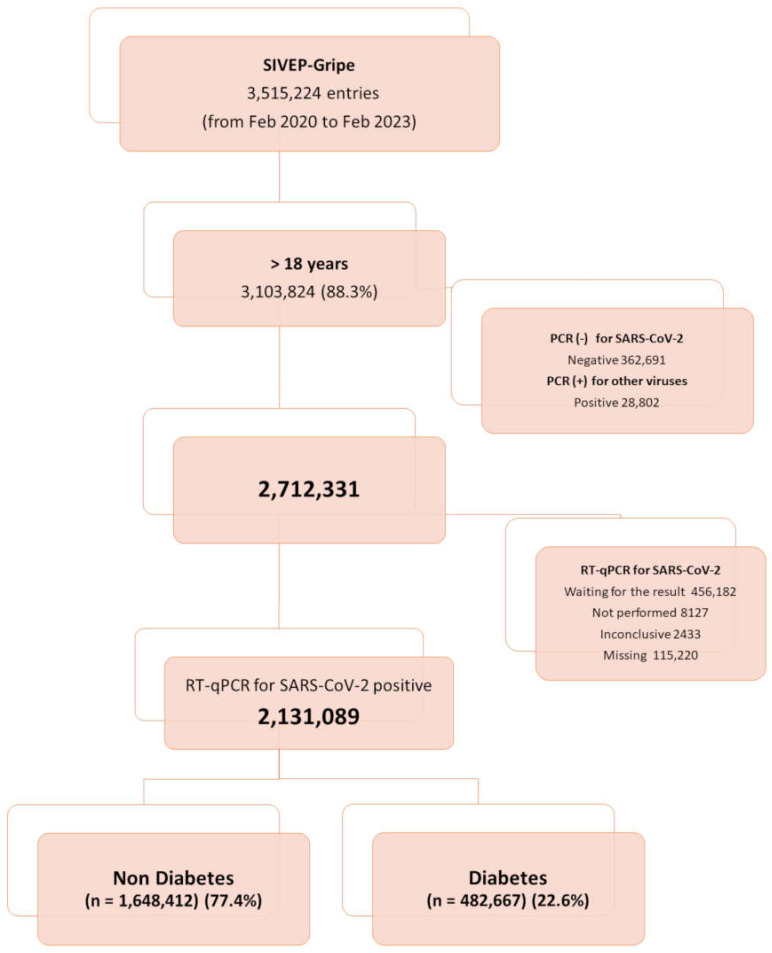
Flowchart of cohort selection.

**Figure 2 microorganisms-13-00979-f002:**
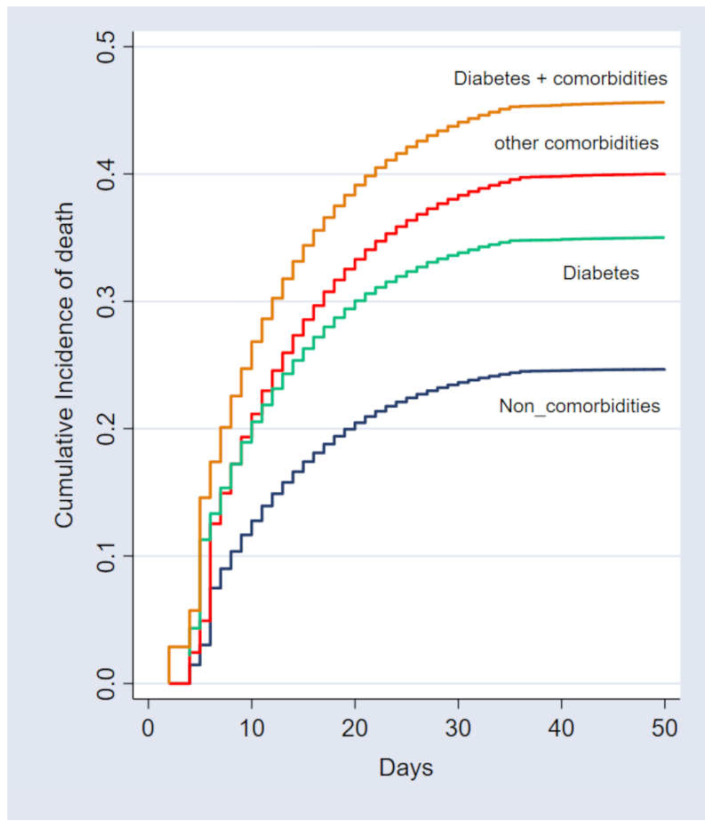
Cumulative incidence of death according to the presence of diabetes mellitus and comorbidities. Statistical test: competing-risk survival analysis. Reference category: individuals without comorbidities (*p* < 0.001).

**Figure 3 microorganisms-13-00979-f003:**
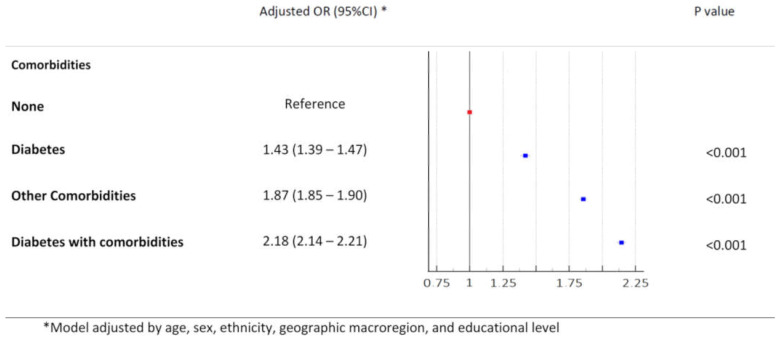
Adjusted odds ratio of death according to the presence diabetes mellitus and other comorbidities. Statistical test: binary logistic model. Reference category: individuals without comorbidities.

**Figure 4 microorganisms-13-00979-f004:**
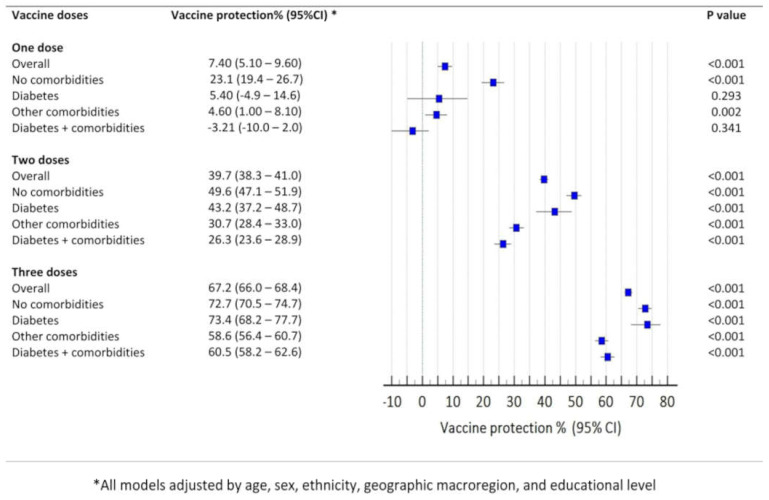
Effectiveness of the vaccine against COVID-19-related deaths according to the presence of diabetes mellitus and other comorbidities. Statistical test: binary logistic models. Reference category: individuals with no dose of the vaccines. All models were adjusted for age, sex, ethnicity, region, educational level, and epidemiological week of hospitalization.

**Figure 5 microorganisms-13-00979-f005:**
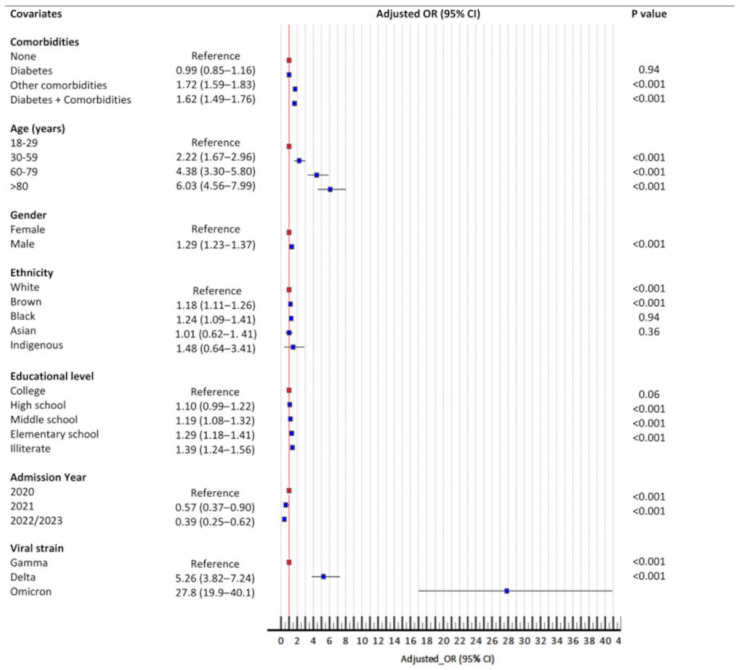
Risk factors of booster failure. Statistical test: binary logistic model.

**Table 1 microorganisms-13-00979-t001:** Demographic and clinical characteristics of hospitalized patients with laboratory-confirmed COVID-19, stratified according to the presence of diabetes mellitus (DM) and other comorbidities (*n* = 2,131,089).

Covariates ^a^	Group 1 (%) 911,270 (42.8)	Group 2 (%) 737,142 (34.6)	Group 3 (%) 100,849 (4.7)	Group 4 (%) 381,828 (17.9)
Age (years)				
Median (IQR)	51.83 (40.1–65.25)	64.6 (51.66–76.91)	61.6 (51.58–71.83)	68.08 (58.83–76.91)
Mean (SD)	53.35 (17.28)	63.71 (17.1)	61.41 (14.76)	67.44 (13.23)
Age group (years)				
18–29.9	69,554 (7.6)	20,811 (2.8)	1945 (1.9)	1862 (0.5)
30–59.9	537,178 (58.9)	279,165 (37.9)	44,252 (43.9)	103,662 (27.1)
60–79.9	228,173 (25)	293,043 (39.8)	43,535 (43.2)	207,714 (54.4)
>80	76,365 (8.4)	144,123 (19.6)	11,117 (11.0)	68,590 (18.0)
Sex (*n* = 2,131,073)				
Male	533,422 (58.5)	393,956 (53.4)	56,109 (55.6)	189,280 (49.6)
Female	377,839 (41.5)	343,183 (46.6)	44,738 (44.4)	192,546 (50.4)
Region				
Southeast	433,950 (47.6)	374,041 (50.7)	50,569 (50.1)	193,660 (50.7)
South	144,818 (15.9)	138,439 (18.8)	13,710 (13.6)	66,197 (17.3)
Central-West	104,479 (11.5)	69,435 (9.4)	9303 (9.2)	32,642 (8.5)
Northeast	152,970 (16.8)	117,259 (15.9)	19,213 (19.1)	69,816 (18.3)
North	75,053 (8.2)	37,968 (5.2)	8054 (8)	19,513 (5.1)
Ethnicity (*n* = 1,739,886)				
White	363,718 (49.8)	334,908 (54.9)	41,753 (49.6)	167,992 (53.3)
Brown	323,001 (44.2)	233,366 (38.3)	36,268 (43.1)	122,952 (39.0)
Black	32,919 (4.5)	33,676 (5.5)	4671 (5.6)	19,673 (6.2)
Asian	9230 (1.3)	6910 (1.1)	1165 (1.4)	3869 (1.2)
Indigenous	2045 (0.3)	1009 (0.2)	244 (0.3)	517 (0.2)
Educational level (*n* = 754,185)				
Illiterate	14,574 (4.7)	20,962 (7.8)	2584 (7.0)	11,999 (8.7)
Elementary	63,500 (20.5)	81,871 (30.3)	11,268 (30.3)	47,969 (34.9)
Middle-School	53,693 (17.4)	51,813 (19.2)	7161 (19.3)	27,769 (20.2)
High-School	120,259 (38.9)	78,782 (29.2)	11,346 (30.5)	34,693 (25.2)
College	57,371 (18.5)	36,629 (13.6)	4793 (12.9)	15,149 (11.0)
Signs/symptoms at presentation				
Fever	525,529 (57.7)	398,280 (54.0)	56,592 (56.1)	198,376 (52.0)
Cough	625,239 (68.6)	499,164 (67.7)	71,337 (70.7)	259,120 (67.9)
Dyspnea	600,330 (65.9)	531,067 (72.0)	70,560 (70.0)	281,396 (73.7)
Odynophagia	186,518 (20.5)	112,622 (15.3)	18,360 (18.2)	56,415 (14.8)
Oxygen saturation <95% (*n* = 1,763,683)	508,688 (70.8)	490,602 (77.7)	63,647 (75.1)	263,152 (79.8)
Number of comorbidities				
None	911,270 (100)	0 (0.0)	0 (0.0)	0 (0.0)
1	0 (0.0)	526,822 (71.5)	100,849 (100)	0 (0.0)
2	0 (0.0)	170,916 (23.2)	0 (0.0)	243,726 (63.8)
3	0 (0.0)	39,404 (5.3)	0 (0.0)	138,102 (36.2)
Major comorbidities				
Cardiology	0 (0.0)	408,461 (55.4)	0 (0.0)	274,329 (71.8)
Hypertension	0 (0.0)	169,165 (23.0)	0 (0.0)	106,296 (27.8)
Obesity	0 (0.0)	125,903 (17.1)	0 (0.0)	59,979 (15.7)
Neurologic	0 (0.0)	76,911 (10.4)	0 (0.0)	32,669 (8.6)
Pulmonary	0 (0.0)	62,110 (8.4)	0 (0.0)	23,492 (6.2)
Renal	0 (0.0)	41,944 (5.7)	0 (0.0)	28,498 (7.5)
Immunosuppression	0 (0.0)	37,762 (5.1)	0 (0.0)	11,088 (2.9)
Oncology	0 (0.0)	29,882 (4.1)	0 (0.0)	7130 (1.9)
Hematology	0 (0.0)	14,675 (2.0)	0 (0.0)	4737 (1.2)
Nosocomial				
No	899,664 (98.7)	718,522 (97.5)	99,327 (98.5)	372,195 (97.5)
Yes	11,606 (1.3)	18,620 (2.5)	1522 (1.5)	9633 (2.5)
SARS-CoV-2 strain				
Ancestral (predominant in 2020)	276,520 (30.3)	38,379 (38.1)	242,250 (32,9)	139,114 (36,4)
Gamma (more prevalent in 2021)	509,287 (55.9)	47,994 (47.6)	351,679 (47,7)	170,823 (44,7)
Delta (more prevalent in 2021)	45,467 (5.0)	4988 (4.9)	41,267 (5,6)	23,313 (6,1)
Omicron (predominant in 2022 and 2023)	79,996 (8.8)	9488 (9.4)	101,946 (13,8)	48,578 (12,7)
Admission Year				
2020	276,520 (30.3)	242,250 (32.9)	38,379 (38.1)	139,114 (36.4)
2021	559,932 (61.4)	397,527 (53.9)	53,695 (53.2)	196,728 (51.5)
2022/2023	74,818 (8.2)	97,365 (13.2)	8775 (8.7)	45,986 (12.0)
Vaccine doses (*n* = 1,887,890)				
None	664,759 (72.9)	490,768 (66.6)	72,370 (80.1)	251,177 (73.6)
One	38,292 (4.2)	38,930 (5.3)	5126 (5.7)	21,426 (6.3)
Two	64,008 (7.0)	81,028 (11.0)	8731 (9.7)	45,759 (13.4)
Three	32,684 (3.6)	45,788 (6.2)	4107 (4.5)	22,937 (6.7)
ICU (*n* = 1,836,099)				
No	518,821 (69.7)	386,078 (58.5)	56,054 (64.2)	185,766 (53.9)
Yes	225,110 (30.3)	274,360 (41.5)	31,235 (35.8)	158,675 (46.1)
Ventilatory support (*n* = 1,809,738)				
None	193,705 (26.5)	120,912 (18.5)	18,355 (21.3)	53,953 (15.9)
Non-invasive	426,169 (58.3)	381,401 (58.5)	50,857 (59.0)	194,575 (57.3)
Invasive	111,639 (15.3)	149,846 (23)	17,045 (19.8)	91,281 (26.9)
Mortality rate				
No	705,799 (77.5)	459,048 (62.3)	67,905 (67.3)	216,656 (56.7)
Yes	205,471 (22.5)	278,094 (37.7)	32,944 (32.7)	165,172 (43.3)

^a^ Data (*n*) in the first column represent the available data for those variables with missing values. Legend: Group 1, patients without comorbidities; Group 2, patients with non-DM comorbidities; Group 3, patients with DM without other comorbidities; and Group 4, patients with DM associated with other comorbidities. For intergroup comparisons, Student’s *t*-test and Pearson’s chi-square test were used, where applicable. All comparisons among the groups were *p* < 0.001.

## Data Availability

All SIVEP-Gripe data are publicly available at the following address https://opendatasus.saude.gov.br/dataset/. Dataset accessed on 20 February 2023. Our analysis code is available upon request from the corresponding author (Eduardo A. Oliveira, eduolive812@gmail.com).

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
