# Peer review of "COVID-19 Vaccine Effectiveness and Risk Factors of Booster Failure in 480,000 Patients with Diabetes Mellitus: A Population-Based Cohort Study"

_microorganisms, 2025, doi:10.3390/microorganisms13050979_

Round 1
Reviewer 1 Report
Comments and Suggestions for Authors
Dear authors,
I have now completed the review of the manuscript titled "COVID-19 vaccine effectiveness and risk factors of booster failure in 480,000 patients with diabetes mellitus: A population-based cohort study."
The manuscript is interesting and, in general, fairly well-written. The study analyzed a large cohort of 2,131,089 patients, including 482,677 with diabetes mellitus (22.6%), providing substantial statistical power for the analyses conducted. The researchers utilized SIVEP-Gripe, a nationwide Brazilian surveillance database tracking severe acute respiratory infections, offering a robust real-world dataset spanning February 2020 to February 2023.
However, I still have some suggestions to further improve the quality of the manuscript.
I would like to suggest that the authors address these limitations in the article, either by discussing them in the limitations section or, where feasible, by making the appropriate revisions:
1. The study focused exclusively on hospitalized patients, potentially skewing results toward more severe cases and limiting generalizability to the broader diabetic population with COVID-19 infections. The absence of individual-level vaccination data (including timing between doses, vaccine types, and antibody responses) limited the researchers' ability to explore waning immunity or vaccine-specific effectiveness.
2. Some important recent findings could be stated in introduction. For example, Immunogenicity of COVID-19 Vaccines in Patients with Diverse Health Conditions: a Comprehensive Systematic Review is directly relevant to the vaccine effectiveness questions raised in the paper, especially regarding patients with comorbidities.
3. The adjusted odds ratio of 1.43 (95% CI, 1.39-1.47) for death in diabetes patients aligns with previous meta-analyses, supporting the reliability of this finding. However, the differential risk between DM alone versus DM with comorbidities (OR 2.18) deserves greater emphasis, as it suggests that diabetes itself may not be the primary driver of poor outcomes. The finding that vaccine effectiveness against death was similar between patients without comorbidities (72.7%) and those with DM alone (73.4%) is encouraging. However, the reduced effectiveness in patients with multiple comorbidities (60.5%) suggests potential biological differences in immune response that warrant further investigation. The dramatic increase in booster failure risk during the Omicron period (aOR = 27.8) overshadows all other risk factors, highlighting the evolving viral landscape as perhaps the most critical determinant of vaccine protection.
4. Discussion would be extended by breifly mentioning latest research, to show readers future research possibilities. For example, An Adaptive Ensemble Deep Learning Framework for Reliable Detection of Pandemic Patients is relevant for understanding machine learning approaches to COVID-19 risk stratification, which could complement the logistic regression models used in the paper. Also, Baseline physical activity is associated with reduced mortality and disease outcomes in COVID-19: A systematic review and meta-analysis relates to modifiable risk factors that might affect COVID-19 outcomes in high-risk populations like diabetic patients.
Thank you for your valuable contributions to our field of research. I look forward to receiving the revised manuscript.
Author Response
General Comments:
Dear authors,
I have now completed the review of the manuscript titled "COVID-19 vaccine effectiveness and risk factors of booster failure in 480,000 patients with diabetes mellitus: A population-based cohort study. The manuscript is interesting and, in general, fairly well-written. The study analyzed a large cohort of 2,131,089 patients, including 482,677 with diabetes mellitus (22.6%), providing substantial statistical power for the analyses conducted. The researchers utilized SIVEP-Gripe, a nationwide Brazilian surveillance database tracking severe acute respiratory infections, offering a robust real-world dataset spanning February 2020 to February 2023.
However, I still have some suggestions to further improve the quality of the manuscript. I would like to suggest that the authors address these limitations in the article, either by discussing them in the limitations section or, where feasible, by making the appropriate revisions:
Answer: The authors express their gratitude for the positive feedback provided by Reviewer #1 regarding our study.
Comment 1. The study focused exclusively on hospitalized patients, potentially skewing results toward more severe cases and limiting generalizability to the broader diabetic population with COVID-19 infections. The absence of individual-level vaccination data (including timing between doses, vaccine types, and antibody responses) limited the researchers' ability to explore waning immunity or vaccine-specific effectiveness.
Answer: We agree with the criticisms and recommendations. In fact, regarding the generalizability of our study, we have already pointed out this limitation in the original manuscript (Page 12, lines 327-328). However, we have highlighted the important limitation pointed out by reviewer #1 regarding the absence of individual-level vaccination data (including timing between doses, vaccine types, and antibody responses). We included the following statement in the Discussion section “Third, the lack of individual-level vaccination data, including the timing between doses, vaccine types, and antibody responses, has limited our capacity to investigate important issues such as waning immunity or vaccine-specific effectiveness” (page 12, lines 328-331).
Comment 2. Some important recent findings could be stated in introduction. For example, Immunogenicity of COVID-19 Vaccines in Patients with Diverse Health Conditions: a Comprehensive Systematic Review is directly relevant to the vaccine effectiveness questions raised in the paper, especially regarding patients with comorbidities.
Answer: We thank the reviewer for bringing our attention to this relevant study. In this regard, we included a brief comment about the vaccine effectiveness raised in the paper, especially regarding patients with comorbidities (Page 2, lines 58-60).
Comment 3. The adjusted odds ratio of 1.43 (95% CI, 1.39-1.47) for death in diabetes patients aligns with previous meta-analyses, supporting the reliability of this finding. However, the differential risk between DM alone versus DM with comorbidities (OR 2.18) deserves greater emphasis, as it suggests that diabetes itself may not be the primary driver of poor outcomes. The finding that vaccine effectiveness against death was similar between patients without comorbidities (72.7%) and those with DM alone (73.4%) is encouraging. However, the reduced effectiveness in patients with multiple comorbidities (60.5%) suggest potential biological differences in immune response that warrant further investigation. The dramatic increase in booster failure risk during the Omicron period (aOR = 27.8) overshadows all other risk factors, highlighting the evolving viral landscape as perhaps
the most critical determinant of vaccine protection.
Answer: We thank reviewer #1 for the insightful comments regarding our findings. Accordingly, we have included two brief comments in the revised Conclusions section about the crucial points raised by reviewer # 1. (See on page 12, lines 349-351 and lines 353-354).
Comment 4. Discussion would be extended by briefly mentioning latest research, to show readers future research possibilities. For example, An Adaptive Ensemble Deep Learning Framework for Reliable Detection of Pandemic Patients is relevant for understanding machine learning approaches to COVID-19 risk stratification, which could complement the logistic regression models used in the paper. Also, Baseline physical activity is associated with reduced mortality and disease outcomes in COVID-19: A systematic review and meta-analysis relates to modifiable risk factors that might affect COVID-19 outcomes in high-risk populations like diabetic patients.
Answer: We agree with the reviewer's comments. Consequently, a concise paragraph has been incorporated to address the public health implications and potential future research directions concerning chronic diseases and the outcomes of COVID-19 1 (page 12, lines 336-342).
Reviewer 2 Report
Comments and Suggestions for Authors
This is a well written and interesting study.
I have some suggestions for the authors:
- In Introduction, the description of the mechanisms through which individuals with diabetes mellitus develop more severe forms of COVID-19 is very brief. I suggest to extend it.
Some studies have shown that patients with diabetes mellitus are at risk of developing severe COVID-19 forms, but they are also more likely to have a prolonged recovery: Tudoran, C.; Tudoran, M.; Cut, T.G.; Lazureanu, V.E.; Bende, F.; Fofiu, R.; Enache, A.; Pescariu, S.A.; Novacescu, D. The Impact of Metabolic Syndrome and Obesity on the Evolution of Diastolic Dysfunction in Apparently Healthy Patients Suffering from Post-COVID-19 Syndrome. Biomedicines 2022, 10, 1519. https://doi.org/10.3390/biomedicines10071519
- I consider that the study would benefit, if the authors would explain in Introduction the presumed mechanisms through which vaccination may protect diabetic patients against severe forms of COVID-19.
- The study would benefit if the authors could provide the extend of the initial lung injury or determine the severity of the COVID-19 initial infection.
- Were patients with acute cardiovascular and respiratory pathologies excluded from the study?
- At line 101, the authors mention that they categorized the co-morbidities by number. I recommend to divide the co-morbidities by type, as patients with cardiovascular and pulmonary pathologies have the highest risk of developing severe forms of COVID-19.
- At line 104, the authors should mention the type of vaccine.
- As the primary goal of this study was to determine the impact of DM on the outcome of patients infected with the SARS-CoV-2 virus, to highlight the statistical significance of their findings, the authors should calculate p-values between 4 patients' groups, or at least between the group of patients with DM and the other 3 groups. Perhaps, to make the table easier to understand, the authors could name A - 911,270 (42.8%) patients 141 without comorbidities, B- 737,142 (34.6%) patients with non-DM comorbidities, etc.
- In Figure 2, the authors should mention the statistical test they used. I recommend to mention the statistical test for each table or figure. The authors can mention it in the Legend or title.
- In Figure 2, was the Cumulative incidence of death statistically significant in patients with DM comparted to those without DM?
- Line 197, how was the degree of vaccine protection established?
- At line 217 the authors mention the predominant reason for booster failure the presumed viral variant. They should mention in Table 1 the number of patients infected in the delta-period or omicron- period, etc. As the variants seem to bee very important, the authors should try to establish each period and the number of patients infected. I understand that it was impossible to determine the variant in each patient, but the authors could try to specify the time interval in which each variant was predominant.
- Lines 283-286 should be mentioned in Results.
Author Response
Comment 1. In Introduction, the description of the mechanisms through which individuals with diabetes mellitus develops more severe forms of COVID-19 is very brief. I suggest extending it.
Some studies have shown that patients with diabetes mellitus are at risk of developing severe
COVID-19 forms, but they are also more likely to have a prolonged recovery: Tudoran, C.; Tudoran, M.; Cut, T.G.; Lazureanu, V.E.; Bende, F.; Fofiu, R.; Enache, A.; Pescariu, S.A.; Novacescu, D. The Impact of Metabolic Syndrome and Obesity on the Evolution of Diastolic Dysfunction in Apparently Healthy Patients Suffering from Post- COVID-19 Syndrome. Biomedicines 2022, 10, 1519.
https://doi.org/10.3390/biomedicines10071519
Answer: We agree with this suggestion and have consequently incorporated concise statements regarding the mechanisms by which individuals with diabetes mellitus may develop more severe manifestations of COVID-19 and potentially experience an extended recovery period (page 2, lines 46-50).
Comment 2. I consider that the study would benefit, if the authors would explain in Introduction the presumed mechanisms through which vaccination may protect diabetic patients against severe forms of COVID-19.
Answer: We agree with this comment. Accordingly, we have added a brief comment and pertinent reference regarding the mechanisms by which vaccination may protect patients with diabetes against severe forms of COVID-19. (page 2, lines 50-52).
Comment 3. The study would benefit if the authors could provide the extend of the initial lung injury or determine the severity of the COVID-19 initial infection.
Answer: This is an important and pertinent point. However, the administrative nature of the dataset employed in this study precluded the inclusion of the clinical details of the individuals, as this information was not available in the SIVEP-Gripe. To partially address this limitation, this study incorporated oxygen saturation at admission as a proxy for the severity of lung injury at baseline in all models. Nonetheless, we have included this issue as another limitation of our study in the revised Discussion section (page 12, Lines 325-327).
Comment 4. Were patients with acute cardiovascular and respiratory pathologies excluded from the study?
Answer: This is an interesting observation. Nevertheless, as pointed out in the previous comment, SIVEP-Gripe is an administrative database of the Ministry of Health created for the surveillance of hospitalized cases of respiratory viral syndrome in Brazil. Unfortunately, the dataset did not include detailed information regarding the clinical condition of the patients.
Comment 5. At line 101, the authors mention that they categorized the co-morbidities by number. I recommend to divide the co-morbidities by type, as patients with cardiovascular and pulmonary pathologies have the highest risk of developing severe forms of COVID-19.
Answer: This is an interesting issue. However, it is evident from Table 1 that cardiology and hypertension are the predominant pre-existing chronic conditions among patients with diabetes, as expected. These findings align with the inquiries posed by Reviewer #2. In the preliminary analyses, models were tested with each comorbidity considered separately. However, due to the involvement of multiple chronic diseases, the models demonstrated reduced robustness.
Comment 6. At line 104, the authors should mention the type of vaccine.
Answer: We agree with this suggestion. Accordingly, we have included this information to the Methods section (page 4, lines 114-118).
Comment 7. As the primary goal of this study was to determine the impact of DM on the outcome of patients infected with the SARS-CoV-2 virus, to highlight the statistical significance of their findings, the authors should calculate p-values between 4 patients' groups, or at least between the group of patients with DM and the other 3 groups. Perhaps, to make the table easier to understand, the authors could name A - 911,270 (42.8%) patients 141 without comorbidities, B- 737,142 (34.6%) patients with non-DM comorbidities, etc.
Answer: We agree with this important suggestion make the table easier to understand. Accordingly, we have modified the Table 1 header and footnote. We have also clarified this information in the Methods section (page 3, lines 87-91).
Comment 8. In Figure 2, the authors should mention the statistical test they used. I recommend mentioning the statistical test for each table or figure. The authors can mention it in the Legend or title.
Answer: We agree with this suggestion. Accordingly, we included the statistical tests used in the figures and table legends.
Comment 9. In Figure 2, was the Cumulative incidence of death statistically significant in patients with DM comparted to those without DM?
Answer: We thank the reviewer for the opportunity to clarify this point. Mortality was evaluated by competing risk analysis using the cumulative incidence function (CIF). Discharge was analyzed as a competing event by competing risk analysis. Therefore, we compared the differences in the cumulative incidence of death between the groups. We have included this information in the Methods section (page 4, lines 136-138) and in Figure 2 legend.
Comment 10. Line 197, how was the degree of vaccine protection established?
Answer: As stated in the Methods section, we used binary regression models to estimate vaccine effectiveness against death conferred by the vaccines among the groups. For this analysis, we developed five models using the binary regression logistic method, including one overall model for the entire cohort and one model for each subgroup included in the analysis. In the models, death was included as a dependent variable, and vaccination status (stratified according to the number of doses) was included as an independent variable. All models were also adjusted by age, sex, ethnic group, geographic macroregions, educational level, viral strain, and epidemiological week of admission. The results are expressed as VE (%) using the formula 100 × (1–adjusted odds ratio) and their 95% confidence intervals (CI).
Nonetheless, we rephrased these statements in the Methods section to clarify this pivotal point raised by reviewer # 2. (Page 4, lines 138-145)
Comment 11. At line 217 the authors mention the predominant reason for booster failure the presumed viral variant. They should mention in Table 1 the number of patients infected in the delta-period or omicron- period, etc. As the variants seem to be very important, the authors should try to establish each period and the number of patients infected. I understand that it was impossible to determine the variant in each patient, but the authors could try to specify the time interval in which each variant was predominant.
The reviewer raised a significant point. Table 1 presents the distribution of the viral strains across the groups, revealing a distribution pattern that closely aligns with the patients' year of admission. According to Brazilian epidemiological data, the ancestral strain was predominant during the first year of the pandemic, whereas the gamma and delta variants were more prevalent in 2021. At the beginning of 2022, there was a notable surge in omicron variant, which subsequently became the predominant strain in 2022 and 2023. We have included in revised Table 1 the information regarding the year for each viral strain.
Comment 12. Lines 283-286 should be mentioned in Results.
Answer: We agree with the suggestion. Therefore, we added this information to the Results section (page 6, lines 182-183).
Round 2
Reviewer 1 Report
Comments and Suggestions for Authors
All comments have been thoroughly addressed. I extend my gratitude to both the authors and editors for taking my opinions into consideration during the review of this manuscript.
Reviewer 2 Report
Comments and Suggestions for Authors
The authors answered all my questions.